# OpenReview forum: "MAVEN-T: Breaking the Imitation Ceiling in Trajectory Prediction with Reinforced Distillation"
_ICLR.cc/2026/Conference — ICLR 2026 Conference Withdrawn Submission_

### Official Review · Reviewer_D1nf · 2025-10-27

**Soundness:** 2
**Presentation:** 1
**Contribution:** 1
**Rating:** 2
**Confidence:** 4

**Summary:**

This paper proposes a reinforcement-augmented distillation framework that enables a student model to surpass its teacher. Specifically, the student actively interacts with its environment and uses reinforcement learning to refine the teacher's knowledge, directly optimizing for task-specific objectives. Consequently, the student achieves a 6.2x parameter compression and a 3.7x inference speedup while developing more robust policies than the teacher.

**Strengths:**

Knowledge distillation is a critical research area in autonomous driving.

**Weaknesses:**

1. The manuscript has a number of problems, such as a lack of citations in the main text (except for the related works section), garbled characters, and inconsistent superscripts/subscripts in the formulas.

2. ​​Lack of Motivation and Ablation Analysis:​​ The integration of RL and distillation seems disjointed. The paper does not sufficiently justify why RL is necessary on top of distillation, as opposed to using RL for training from scratch. Furthermore, a critical ablation study is missing to demonstrate that the synergistic effect of both components is essential for the achieved performance, rather than each one working in isolation.

3. The structure of the model seems to be a combination of various existing methods, but it takes up a lot of space in the article, which is not the main innovation. In contrast, the core rl module is only briefly introduced.

**Questions:**

See weaknesses

---

### Official Review · Reviewer_3zcH · 2025-10-31

**Soundness:** 3
**Presentation:** 2
**Contribution:** 2
**Rating:** 2
**Confidence:** 4

**Summary:**

This paper proposes a reinforcement-augmented knowledge distillation framework to overcome the “imitation ceiling” in knowledge distillation and model compression, where a distilled student model can only match but not surpass its teacher. The method combines multi-level distillation with reinforcement learning to refine and improve the student’s policy beyond pure imitation. A high-capacity teacher model guides a lightweight GRU-based student model for trajectory prediction task. The training uses a progressive adaptive curriculum that gradually increases scenario complexity, along with a multi-granular loss aligning features, attention, and semantics. The student further fine-tunes its policy via reinforcement learning using safety, comfort, and efficiency rewards.

**Strengths:**

The combination of knowledge distillation and reinforcement learning to go beyond pure imitation is a reasonable idea.

**Weaknesses:**

1. The proposed framework would be more reasonable if applied to learning a control policy or planner from data rather than to trajectory prediction. The learning objective for trajectory prediction is to approximate the distribution of human-driven vehicle trajectories, whereas reinforcement learning (RL) optimizes a policy to maximize expected rewards. Maximizing a set of hand-crafted rewards does not necessarily lead to accurate trajectory distribution modeling. The authors should therefore provide a clear justification for why the proposed RL-based framework is suitable for trajectory prediction. Although the proposed model achieves good empirical performance compared to baselines, further analysis is needed to demonstrate that the RL component actually contributes to the improvement. In particular, the authors should compare the prediction accuracy of the teacher and student networks under the same evaluation metrics. Currently, the results are reported using different metrics (RMSE vs. ADE/FDE), which makes them not directly comparable. There appears to be no clear reason why different metrics must be used. An ablation study isolating the effect of the RL loss on the distillation performance would also strengthen the paper’s claims.
2. The datasets (e.g., NGSIM) and baselines used in the experiments are a bit outdated. The authors are encouraged to evaluate their model on more challenging datasets that capture complex urban driving scenarios (e.g., Waymo Open Motion Dataset) and compare against state-of-the-art trajectory prediction models on those benchmarks.
3. The paper was written somewhat carelessly, with several missing references (e.g., EWC, baseline methods) and uncommented Latex code on Page 5. The authors should carefully revise the paper to ensure completeness and clarity of citations and improve overall writing quality.

**Questions:**

How is the reward function defined? In particular, how is the goal specified in the reward function?

---

### Official Review · Reviewer_ep8n · 2025-11-01

**Soundness:** 2
**Presentation:** 2
**Contribution:** 3
**Rating:** 4
**Confidence:** 3

**Summary:**

- The paper proposes a method to perform reinforcement learning (RL) augmented knowledge distillation to outperform the teacher’s performance in the case of multi-agent trajectory prediction task.
  - The student, instead of ‘imitating’ the teacher via simple supervised learning loss, would try to refine its policy using RL to correct the teacher's distilled knowledge and thereby achieve better and robust optimal policies than the teacher.
- The authors try to combine the high representative and reasoning power of the teacher and progressively distill that into small deployment-ready models.
- The authors introduce MAVEN-T (Multi-Agent enVironment-aware Enhanced Neural Trajectory) that introduces complementary architectures for teacher and student, RL based progressive distillation and multi-granular knowledge distillation.
- The paper was able to achieve at par accuracy with 6.2x compression and 3.7x inference speedup.

Given the comments related to weaknesses and limitations, I can have the rating as 4. Flexible to move this during or after rebuttal.

**Strengths:**

- The paper proposes a methodology to try to handle the trade-off between computational complexity and better reasoning and representation capacity.
- The authors evaluated how the student is better in robustness to noise than the teacher that it distilled its knowledge from.
  - This can mean that the student could have explored the state space and that is not there in the static teacher’s dataset.
- The combination of imitation loss, RL loss and the regularizer helps the student learn robust and better policies.
- The authors have done multiple ablations to study the effect of different components of the architecture and various distillation strategies.
- Approaches like these can help distill the capabilities of bigger representative models into smaller and deployable systems with minimal to no compromise in performance.

**Weaknesses:**

1. The equations and the figures can be made a bit more clear.

2. The formatting of the text can be improved. References to appropriate concepts and/or architectures can be included.

3. The performance comparison between student and teacher on the basis of safety and comfort metrics seems to be missing.

4. ADE/FDE metric comparison based on time-horizon of teacher-student seems to be missing.

5. RMSE metric comparison based on time-horizon for teacher-student seems to be missing.

**Limitations**

- It would be great to test OOD generalization performance for the distilled student as well.

- The framework proposed seems to be too much tied to the teacher and student’s architecture. It would be good to have an architecture agnostic study as well.

**Questions:**

1. Line 042: Is it “Mamba”? Can we please add the reference to it as well whatever that concept/architecture is?

2. Figure 1:
  - The flow of the input to the surround-aware encoder was a bit unclear. Can we please redesign the figures with proper input flow?

  - What is AKDM (mentioned on the right side of the figure)?

3. Equation 1:
  - What is $E_{t}$ and $E_{t}^{Hybrid}$ ?

4. Equation 2:

 - What is $E_{t}^{GRU}$ ?

5. Equation 4:

  - I assume there should be a subscript for ht and other terms?

6. Was history included in the observation space to predict the trajectory?

7. Were any on-device deployment tried and were there any measures related to latency of generated outputs?

8. Were there analysis or comparison done with [SceneTransformer architecture](https://arxiv.org/abs/2106.08417) (also has different attention layers for temporal and spatial understanding).


**Suggestion:**

1. It would be great to include the description of each term in the equations for better understanding and reference.

2. Equation 5:

  - Please mention what is $i$ and $E_{i}()$

3. Equation 6:

  - Subscript for the loss components would be good.

4. Line 265: There is a typing error while mentioning algorithm number (it “is Algorithm ??” for now)

5. Line after 269: please remove unnecessary text.

6. Please include line numbers in Algorithm 1.

7. Fig 3,4,5: Can we please have legends for color coding for other vehicles and trajectories in the figures.
It’s appreciated that the authors have provided the information related to which teacher-student features were matched in the appendix, it would be great to refer the readers to the appendix at such instances.

---

### Official Review · Reviewer_Ly6h · 2025-11-02

**Soundness:** 2
**Presentation:** 2
**Contribution:** 1
**Rating:** 2
**Confidence:** 4

**Summary:**

This paper presents MAVEN-T, a teacher-student framework for training low-capacity trajectory prediction models. It features 1) largely different architecture designs for the teacher and the student and 2) curriculum that increases scenario's complexity based on student's performance and 3) RL objective for the student

**Strengths:**

+ The paper is well written
+ Comprehensive ablation on architecture variations

**Weaknesses:**

- While the results in table 2/3 are compelling, they do not provide evidence that the presented approach is capable of training a smaller / more efficient network with equal or better performance. A pareto front comparison would have suffice here, with the presented approach capable to moving the pareto front outward compared to baselines.
- The presented approach is evaluated only on niche datasets (NGSIM / highD), whereas there are well-calibrated public datasets for trajectory prediction (argoverse / WOMD) available. Can the authors evaluated on these datasets so readers would understand how calibrated the presented approach is?

**Questions:**

Similar to the weaknesses section above.
- Why does table 1 and table 2 evaluate on different main metrics (RMSE vs ADE/FDE)? I highly recommend the authors to put up a combined table of baselines and teachers/students along side with their metrics and latency. Otherwise it's hard for readers to have a direct idea on the performance
- Can the authors

---

### Note · Authors · 2025-11-12

I have read and agree with the venue's withdrawal policy on behalf of myself and my co-authors.